# Tribological Properties of Mo-Si-B Alloys Doped with La_2_O_3_ and Tested at 293–1173 K

**DOI:** 10.3390/ma12122011

**Published:** 2019-06-23

**Authors:** Wenhu Li, Taotao Ai, Hongfeng Dong, Guojun Zhang

**Affiliations:** 1School of Materials Science & Engineering, Xi’an University of Technology, Xi’an 710048, China; zhangguojun@xaut.edu.cn; 2School of Materials Science & Engineering, Shaanxi University of Technology, Hanzhong 723000, China; aitaotao0116@126.com (T.A.); dhf@snut.edu.cn (H.D.)

**Keywords:** Mo-Si-B alloys, liquid-liquid doping, tribological properties, wear-resistant

## Abstract

According to the stoichiometric ratios of Mo-10Si-7B, Mo-12Si-8.5B, Mo-14Si-9.8B, and Mo-25Si-8.5B, some new Mo-Si-B alloys doped with 0.3 wt % lanthanum (III) oxide (La_2_O_3_) were prepared via liquid-liquid (L-L) doping, mechanical alloying (MA), and hot-pressing (HP) sintering technology. The phase-composition and microstructure were investigated by X-ray diffraction (XRD) and scanning electron microscope (SEM). The worn surfaces of the plate specimens were studied by confocal laser scanning microscopy (CLSM). Then, the tribological properties of Mo-Si-B alloy doped with sliding plate specimens of 0.3 wt % La_2_O_3_ were investigated against the Si_3_N_4_ ball specimens. The friction coefficients of Mo-Si-B alloys decreased and the wear rates of the alloys increased with test load. The high-temperature friction and wear behavior of Mo-Si-B alloy are related to the surface-oxidation and contact-deformation of the alloy at a high temperature. The low friction coefficients and the reduced wear rates are thought to be due to the formation of low friction MoO_3_ films. MoO_3_ changed the contact state of the friction pairs and behaved as lubricating films.

## 1. Introduction

As a new type of super-alloy material, Mo-Si-B is a high-temperature melting alloy with high strength, hardness, and good oxidation resistance, and has bene the focus of many researchers [1,2,3]. In particular, Mo-Si-B alloy is generally composed of α-Mo, Mo_3_Si, Mo_5_Si_3_, Mo_5_SiB_2_ (T2), and so on. Among them, α-Mo has good fracture-toughness, but Mo_3_Si and Mo_5_Si_3_ inter-metallic compounds and Mo_5_SiB_2_ (T2) phase, produced via a substitution-mechanism, are intrinsically brittle compounds [4,5,6]. α-Mo strongly affects the deformation and fracture-behavior of Mo-Si-B alloys. Therefore, the fracture-toughness of the alloy increases with increasing volume fraction of the α-Mo phase. Moreover, the alloy shows higher fracture-toughness when the α-Mo phase is continuously distributed in fine grains rather than in dispersed particles. The Mo_5_SiB_2_ and Mo_3_Si phases have excellent oxidation resistance at high temperatures as well as a high melting point of >2000 °C. However, these phases have a very low fracture-toughness of about 2–3 MPa·m^1/2^ at room temperature [7,8,9].

Several investigations have discussed the mechanical properties and oxidation resistance of Mo-Si-B alloy. Zhang [10] showed that alloys with Si/B ratios of 2.4 and 1.4 possess a continuous α-Mo matrix with homogeneously distributed inter-metallic particles, whereas the alloy with a Si/B ratio of 0.7 is a homogeneous mixture of α-Mo, Mo_3_Si, and Mo_5_SiB_2_ phases. The Mo_3_Si phase showed the highest hardness and elastic modulus values of the order of 34.9 GPa and 390.4 GPa [11]. The room temperature fracture-toughness of 5–7 MPa·m^1/2^ was observed for 12Si-8.5B alloy [12]. Li [13] reported that Mo-12Si-17B alloy has a large quantity of Mo_5_SiB_2_ phase accompanied by a small quantity of α-Mo phase, which not only promotes rapid formation of the oxide layer but also covers the alloy matrix uniformly within the temperature range of 1000 to 1300 °C. 

To the best of our knowledge, Mo-Si-B alloys are the leading candidates for the next generation of high-temperature structural materials because of their potential to increase the available temperature. Many researches focused on improving the fuel-to-power efficiency of engines including gas turbines [14]. One of the major drawbacks that limits the applications of Mo-Si-B alloys is the poor tribological properties of these alloys even under a small loading. However, the high friction coefficient and difficult processing of this super-alloy has restricted its wide application in industry. At present, the study of the friction and wear properties of Mo and its alloy has become a hot spot in the field of material science and tribology. Deeply exploring and studying the friction and wear mechanisms of Mo-Si-B alloys may provide a scientific basis for the optimization and application of friction-matching Mo-Si-B alloys, which is beneficial for extending the application of Mo-Si-B alloys in many industrial fields. Thus, the main objective of this study was to examine the effect of silicon and boron contents on the micro-structural and tribological properties of Mo-Si-B alloy doped with lanthanum (III) oxide. Mo-Si-B alloys doped with 0.3 wt % La_2_O_3_ were prepared via liquid-liquid (L-L) doping, mechanical-alloying (MA), and hot-pressing (HP) sintering technology. The effects of different Si and B contents on the micro-structure, friction, and wear properties of the alloy were discussed.

## 2. Experimental Procedure

First, ammonium tetra-molybdate solution ((NH_4_)_2_Mo_4_O_13_) and an aqueous solution of lanthanum (III) nitrate (La(NO_3_)_3_) were prepared. Then, (NH_4_)_2_Mo_4_O_13_ was crystallized to form ammonium di-molybdate ((NH_4_)_2_Mo_2_O_7_) in a reaction kettle. Ammonium lanthanum di-molybdate (NH_4_La(Mo_2_O_7_)_2_) was synthesized by the uniform mixing of (NH_4_)_2_Mo_2_O_7_ solution with La(NO_3_)_3_ solution. Thereafter, Mo-La_2_O_3_ powder was obtained by filtrating, drying, calcinating, and reduction with hydrogen. According to the stoichiometric ratios of Mo-10Si-7B, Mo-12Si-8.5B, Mo-14Si-9.8B, and Mo-25Si-8.5B, the main materials were Mo-La_2_O_3_ powder, B, and Si powders for batching. Both Si and B powders were 99.99% pure. Mo-La_2_O_3_, Si, and B powders were mechanically alloyed under a protective atmosphere (argon) with a speed of 250 rpm and a powder:ball weight-ratio of 1:10 for 15 h. After MA, the powders were hot-pressed at 50 MPa and then sintered in vacuum at 1600 °C for 2 h. The compositions of the experimental alloys are listed in Table 1.

The crystalline phases of the alloys were determined by using an X-ray diffraction (XRD, Ultima Ⅳ, Japan science co., Ltd., Tokyo, Japan) equipped with a CuK_α_ radiation source. Micro-structural analyses of the alloys were performed via scanning electron microscopy (SEM) (JSM-6700F, Japan Electronic Co., Ltd., Tokyo, Japan).

The Vickers hardness of the sintered specimens was tested using a HV-1000 Vickers hardness tester (Shanghai caicheng measuring instrument Co., Ltd., Shanghai, China) with a loading force of 300 g and a holding time of 10 s. The average value of 10 points was obtained for each specimen test.

The friction and wear properties of each plate specimen sliding against the Si_3_N_4_ ball specimens were investigated using a high-temperature ball-on-plate tribometer (HT-BPT, GHT-1000E) manufactured by Lanzhou Zhongke Kaihua Technology Development Co., Ltd. (Lanzhou, China) The size and mechanical properties of Si_3_N_4_ balls selected in this experiment are shown in Table 2. The friction and wear testing conditions were: load, 1.96 N, 5.88 N, and 10.78 N; testing time, 30 min; rotating speed, 797.4 rad/min; radius, 10 mm; and testing temperature, 298, 973, 1073, and 1173 K.

After friction and wear tests, a JB-5C surface-profiler (SP) was used to characterize the wear rate of alloys (Shanghai Taiming Optical instrument Co., Ltd., Shanghai, China), which can measure the surface roughness and contour size of the specimen with an optical resolution of up to 0.01 μm. Since a grinding mark without a regular spherical morphology formed during the friction test, the size of the grinding mark profile was measured at different positions. It is impossible to calculate the grinding mark volume using an integral method. In this experiment, the outline of the grinding mark was processed, which was equivalent to a set of discrete data. The dispersion of discrete data determined the calculation accuracy of the wear area. The range between the two adjacent data points is equivalent to a micro-rectangle. The rectangular area is equal to the product of the difference of transverse coordinates (*δx_i_*) and the value of longitudinal coordinate (*y_i_*) of the corresponding position. If we assume that *δx_i_* is equal, the cross-section area of the grinding-mark is as follows:(1)$$A=\sum S_{i}=\sum \delta x_{i}\;y_{i}=\sum \delta x\;y_{i}.$$

The length of the grinding-mark is derived from the circumference of the center-point of the grinding-mark, that is:(2)L=2πr.

Therefore, the volume *V* of the grinding-mark is:(3)$$V=LA=2\pi r\sum \delta \;xy_{i}.$$

## 3. Results and Discussion

### 3.1. XRD Analysis and Micro-Structure Characterization

Figure 1 shows the XRD patterns of different samples of the alloy. The main components of 10Si-7B, 12Si-8.5B, and 14Si-9.8B alloys were α-Mo, Mo_3_Si, and Mo_5_SiB_2_ phases, as well as a small amount of residual mesophase compound (Mo_5_Si_3_). The main components of 25Si-8.5B alloy were Mo_3_Si and Mo_5_SiB_2_ phases, and the peak of α-Mo phase was obviously weakened. This finding indicates that the phase composition of the alloys was not affected by doping with 0.3 wt % La_2_O_3_. Due to the small amount of the doped La_2_O_3_, the corresponding peak value could not be detected by XRD. 

In the process of mechanical alloying, a part of the large impact kinetic energy was converted into the surface energy of the mixed powders. The surface energy of the mixed powder increased after MA. Since MA was conducted in an argon protective environment, Mo, Si, and B powders were not oxidized during MA. Therefore, there was no corresponding characteristic peak for MoO_3_, SiO_2_, and B_2_O_3_.

Representative micrographs of the alloys doped with 0.3 wt % La_2_O_3_ are provided in Figure 2. The bright phase is the α-Mo matrix, which has the highest and the lowest volume fractions in 10Si-7B and 25Si-8.5B alloys, respectively. The other two phases are Mo_3_Si and Mo_5_SiB_2_. They are dispersed within the continuous α-Mo matrix in 10Si-7B and 25Si-8.5B alloys. From Figure 2, the α-Mo phase became discontinuous and the content decreased gradually with increasing silicon and boron contents.

The micro-structure of 14Si-9.8B alloy is characterized by continuous distribution of the inter-metallic compounds and the intermittent distribution of α-Mo phase. The 25Si-8.5B alloy is comprised almost completely of total inter-metallic compound-phase, and the content of α-Mo phase decreased to the least, which was highly consistent with the results of XRD analysis.

### 3.2. Density and Vickers Hardness Analyses

Figure 3 shows the Vickers hardness and density of 10Si-7B, 12Si-8.5B, 14Si-9.8B, and 25Si-8.5B alloy specimens after sintering. The density of Mo-Si-B alloys decreased with the increase in Si and B contents due to the lower density of Si and B than that of Mo, which aligns with the changes in the theoretical density. Furthermore, variations in the density of samples prepared by L-L doping with La_2_O_3_ were similar to those of the Mo-Si-B alloys prepared by solid-solid and solid-liquid doping with La_2_O_3_ [15,16,17]. It has been shown that the doping mode of La_2_O_3_ in the alloy has little effect on the density of the material.

The Vickers hardness of the prepared alloy specimens showed that the hardness of all four alloys was increased gradually with enhancing the Si and B contents. The volume fraction of the inter-metallic compounds, such as Mo_5_SiB_2_, Mo_5_Si_3_, and Mo_3_Si, increased with the increase in Si and B contents. These inter-metallic compounds are formed by in-situ reaction at a high temperature. A certain amount of Si and B enters the lattice of Mo to form an α-Mo based solid solution, which causes some lattice distortion. The increased lattice distortion improves the ability of the material to resist the Vickers diamond head.

### 3.3. Effect of Load on Dry Friction and Wear Properties of Mo-Si-B Alloy

Figure 4 shows the friction coefficients and wear rates of plate specimens of Mo-Si-B alloy sliding against the Si_3_N_4_ ball specimens at 1.96, 5.88, and 10.78 N. As Figure 4a shows, the friction coefficients of the four different alloys decreased with the increase in the test load because the contact surface of the alloy increased in the elastic-plastic state during sliding friction. Thus, there was a non-linear relationship between the actual contact area and loading of the two pairs of the sliding materials. The progressive interaction between the two friction surfaces was promoted by increasing the loading, that is, the surface asperities were polished and the surface roughness decreased. This led to a decrease in the friction coefficient of the alloy with increasing loading. When the load was 1.96 N, the friction coefficients of 10Si-7B, 12Si-8.5B, 14Si-9.8B, and 25Si-8.5B were 2.12, 2.01, 1.98, and 1.25, respectively. The friction coefficients of 10Si-7B, 12Si-8.5B, 14Si-9.8B, and 25Si-8.5B were 1.94, 1.97, 1.81, 1.07 at 5.88 N and 1.58, 1.43, 1.20, 0.75 at 10.78 N, respectively.

Figure 4b shows that the wear rates of the four different alloys increased with the test load. The load affected the actual contact area of the friction pairs and the deformation degree of the alloy under the pressure during sliding friction. Therefore, the real contact area of the friction pairs increased, the deformation degree of the alloy surface increased, and a greater friction stress on the surface of alloy was generated. As a result, more La_2_O_3_ particles detached from the alloy, which increased the three-body abrasion. Thus, further particles of La_2_O_3_ aggravated the wear. Figure 4c shows the wear rates of the paired Si_3_N_4_ ball specimens. The wear rates of Si_3_N_4_ ball specimens increased with the increase in Si and B contents. The increased wear rates are thought to be due to the increased hardness of the Mo-Si-B alloy.

### 3.4. Effect of Temperature on Dry Friction and Wear Properties of Mo-Si-B Alloy

Figure 5a shows the friction coefficients of Mo-Si-B alloy plate specimens sliding against the Si_3_N_4_ ball specimens in the temperature range of 298 to 1173 K and under the experimental load of 5.88 N. With regard to 10Si-7B alloy and 12Si-8.5B alloy doped with 0.3 wt % La_2_O_3_, the coefficients of friction decreased gradually in the temperature range of 298 to 1173 K. The friction coefficients of the 10Si-7B alloy plate specimens were as low as 0.45–0.98 in the temperature range of 973 to 1173 K. However, the steady friction coefficients of the 10Si-7B alloy plate specimens were as high as 1.98 at 298 K. The friction coefficients of 12Si-8.5B alloy were 0.52–0.85 and 1.97 at the test temperature of 973–1173 K and at 298 K, respectively. However, for the alloys of 14Si-9.8B and 25Si-8.5B doped with 0.3 wt % of La_2_O_3_, the friction coefficients decreased with temperature in the range of 298 to 1073 K and increased first at 1173 K. The friction coefficients of 14Si-9.8B alloy were 1.01–1.26 and 1.81 at the test temperature from 973 to 1173 K and at 298 K, respectively. For 14Si-9.8B alloy, the lowest friction coefficient of 1.01 was attained at the test temperature of 1073 K. For 25Si-8.5B alloy, the friction coefficients of 0.88–1.05 and 1.07 were determined at the test temperature from 973 to 1173 K and at 298 K, respectively.

Figure 5b,c shows the wear rates for the plate specimens of Mo-Si-B alloy sliding against the Si_3_N_4_ ball specimens in the temperature range of 298 to 1173 K. The volume wear rates of Mo-Si-B alloy doped with 0.3 wt % La_2_O_3_ decreased. The results showed that the oxide film produced at a high temperature played a significant role in the lubrication and, therefore, the wear rate was effectively reduced. The Si_3_N_4_ ball specimens sliding against the Mo-Si-B alloy plate specimens exhibited the smallest specific wear rates at 1173 K. The smallest specific wear rates at 1173 K are thought to be due to the formation of some low friction films, as described in Figure 5.

Figure 6 shows the friction coefficients of Mo-Si-B alloy doped with 0.3 wt % La_2_O_3_ plate specimens sliding against Si_3_N_4_ ball specimens in the temperature range of 973 to 1173 K. At 973 K, the friction coefficients of Mo-Si-B alloys were relatively high at the beginning of friction test and, then decreased to be stable with prolonging the friction process. When the temperature was 1073 K, the initial friction coefficients of the alloys changed considerably; and simultaneously, the average friction coefficients were lower than those obtained at 973 K. At 1173 K, the lowest average friction coefficients were obtained for 10Si-7B and 12Si-8.5B alloys. However, for 14Si-9.8B and 25Si-8.5B, the average friction coefficients were higher than those attained at 1073 K. The real-time friction coefficient of 25Si-8.5B alloy fluctuated the most.

The high temperature friction and wear behavior of Mo-Si-B alloy should be related to the surface oxidation and contact deformation of the alloy at a high temperature. Figure 7 shows SEM images of 10Si-7B alloys obtained after friction test at 973 K (Figure 7a), 1073 K (Figure 7b), and 1173 K (Figure 7c,d). Figure 7e,f depicts the responding EDS analyses of P1 and P2 in Figure 7d, respectively. According to the corresponding EDS results, the brightest appearing phase particles (labeled as P1) were mainly Mo, Si, O, and La, which indicates that the second phase particles exist in the form of oxide. The existence of MoO_3_, SiO_2_, and La_2_O_3_ can be detected by calculating the content ratio of Mo, Si, and O atoms. The slightly darker gray phase (labeled as P2) are assigned as Mo3Si and oxide.

The friction coefficient depends on the surface contact between the Mo-Si-B alloy and the Si_3_N_4_ ceramic ball. The low friction coefficients were thought to be due to the formation of low friction MoO_3_ films. When the alloy was rubbed at 973 K, molybdenum oxide (MoO_3_) gradually formed on the surface of the alloy. Although MoO_3_ is a volatile compound, it was constantly pressed into the surface of the alloy by the ceramic ball under the pressure and volatilization of MoO_3_ was postponed. MoO_3_ acted as a lubricating film. Therefore, a lower friction coefficient for the alloy was observed at 973 K compared to 298 K. 

When the test temperature was 1173 K, the low melting MoO_3_ (1068 K) and B_2_O_3_ (723 K) began to melt partially to form a liquid phase and the contact state between the friction pairs changed again. For 10Si-7B and 12Si-8.5B alloys, the friction coefficients were further reduced with the increase in friction temperature. Since the viscosity of a liquid phase with a high content of SiO_2_ in 14Si-9.8B and 25Si-8.5B alloys is higher than the low content of SiO_2_ in 10Si-7B and 12Si-8.5B alloys, the liquid phase not only played the role of liquid lubrication but also caused the adhesion of friction pairs and led to an increased friction coefficient. In addition, the high viscosity of the liquid phase could not cover the friction surface evenly and thus resulted in the fluctuation in the friction coefficient of the 25Si-8.5B alloy.

Figure 8 shows the laser confocal microscope images of the worn surfaces on the Mo-Si-B alloy plate specimens after sliding at 298 K. With increasing Si and B contents, the volume fraction of the inter-metallic compounds increased and the content of α-Mo phase with low hardness and good plasticity gradually decreased. The ploughing wear of the alloy surface was reduced by the hard particles as characterized by the decrease in wear volume, the number, and depth of the furrows on the surface. 

According to the laser confocal images of the wear tracks in Figure 8i–l, obvious furrows were observed on the surfaces of Mo-Si alloys, as highlighted by arrows in Figure 8. The depth and width of the furrow on the 10Si-7B alloy were large and the furrows were continuous in the parallel direction and no discontinuity was observed in the middle. With enhancing Si and B contents, the width and depth of the furrow clearly decreased and the number of convex peaks (hard inter-metallic compounds) in the grinding mark increased. However, the wear tracks became shallow and narrow and the shape was no longer continuous and parallel but curved or even interrupted, as highlighted by the rectangles in Figure 8. These phenomena indicate that when the hard abrasive particles encountered more brittle inter-metallic compounds, the ploughing behavior was hindered, the ploughing effect was interrupted, and the wear resistance of the alloy was increased. 

Related research has shown that the hardness is one of the main parameters used to characterize the wear resistance of a material, and in the present study, the wear resistance of Mo-Si-B alloy increased with the increase in the hardness of the material [18,19]. This is consistent with the findings of Archard who found that the wear resistance and hardness of materials are linearly related [20].

## 4. Conclusions

(1) With increasing silicon and boron contents, the volume fraction of inter-metallic compounds increases, and the content of α-Mo phase with low hardness and good plasticity decreases gradually. 

(2) The density of Mo-Si-B alloys decreases, but the hardness of the alloys increases gradually with increasing Si and B contents. 

(3) The friction coefficients of Mo-Si-B alloys decrease and the wear rates of alloys increase with increasing test load. The high-temperature friction and wear behavior of Mo-Si-B alloy is related to the surface oxidation and contact deformation of the alloy at high temperatures. The low friction coefficients are thought to be due to the formation of low friction MoO_3_ films.

## Figures and Tables

**Figure 1 materials-12-02011-f001:**
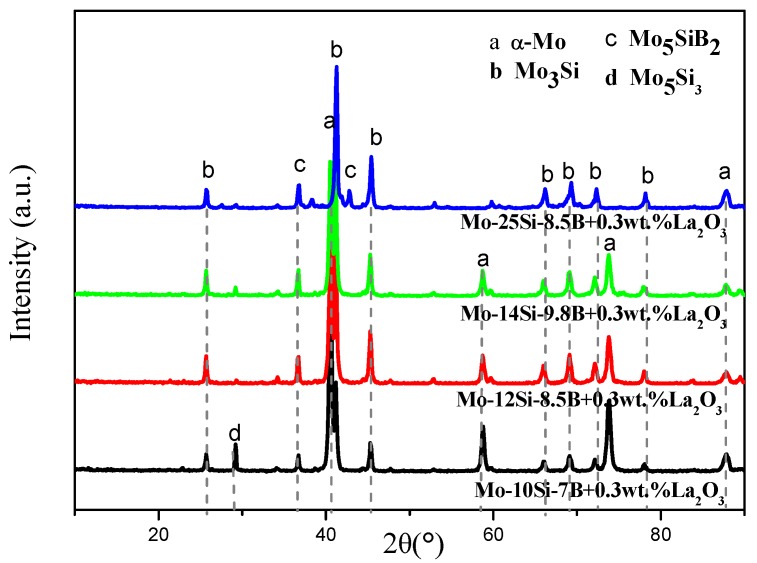
XRD patterns of different Mo-Si-B alloys.

**Figure 2 materials-12-02011-f002:**
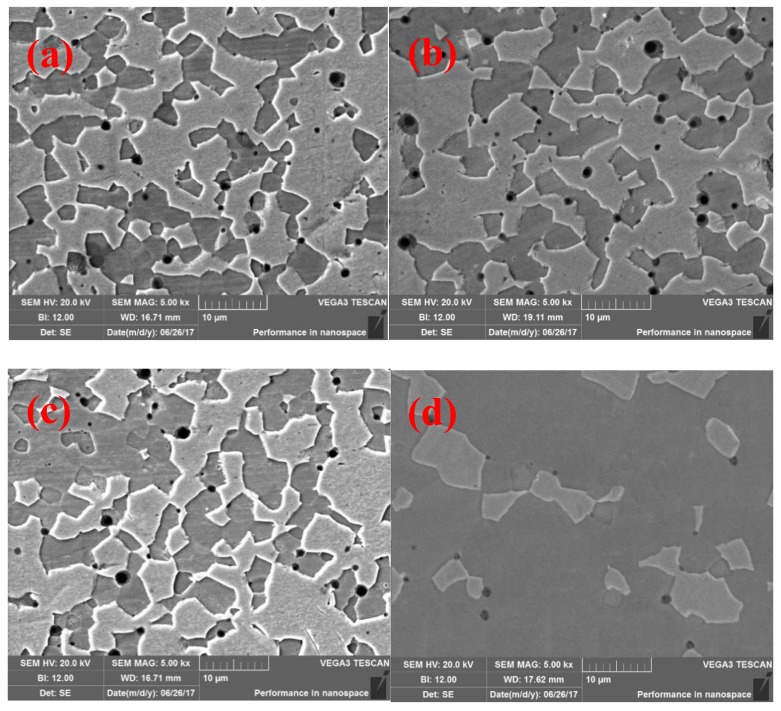
SEM images of the sintered alloys. (**a**) 10Si-7B alloy, (**b**) 12Si-8.5B alloy, (**c**) 14Si-9.8B alloy, and (**d**) 25Si-8.5B alloy.

**Figure 3 materials-12-02011-f003:**
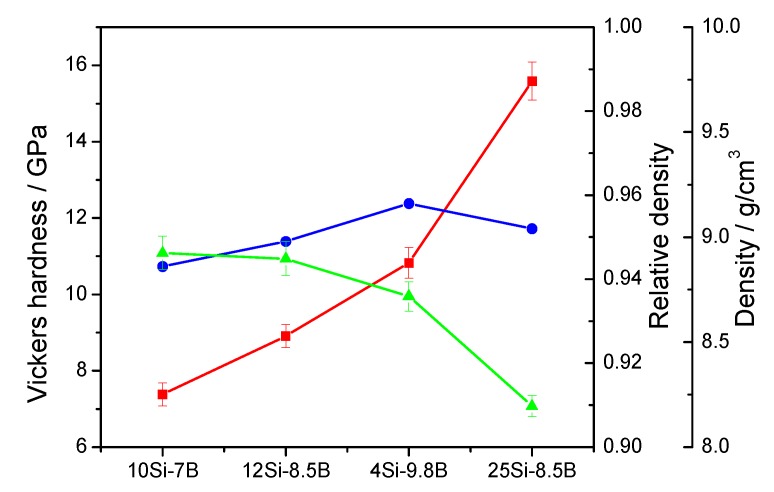
Vickers hardness and density of 10Si-7B, 12Si-8.5B, 14Si-9.8B, and 25Si-8.5B alloys.

**Figure 4 materials-12-02011-f004:**
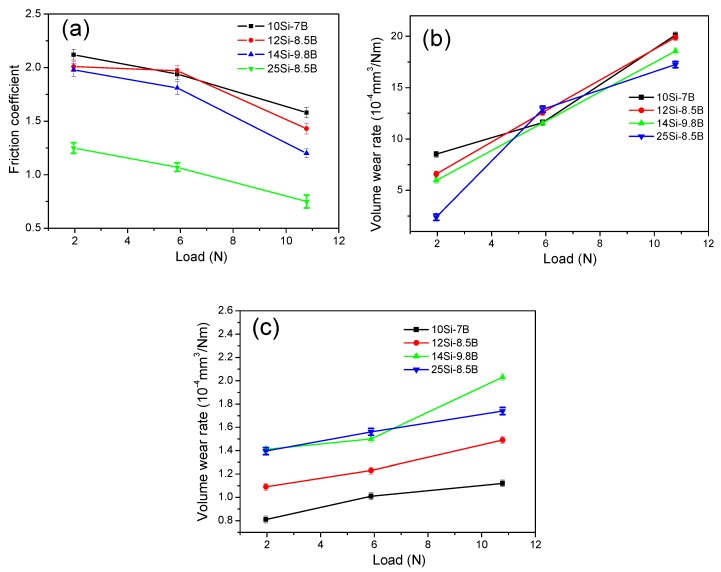
(**a**) Friction coefficients and (**b**) wear rate curves for Mo-Si-B alloy doped with 0.3 wt % La_2_O_3_ plate specimens under different experimental loads and (**c**) wear rates of their paired Si_3_N_4_ ball specimens.

**Figure 5 materials-12-02011-f005:**
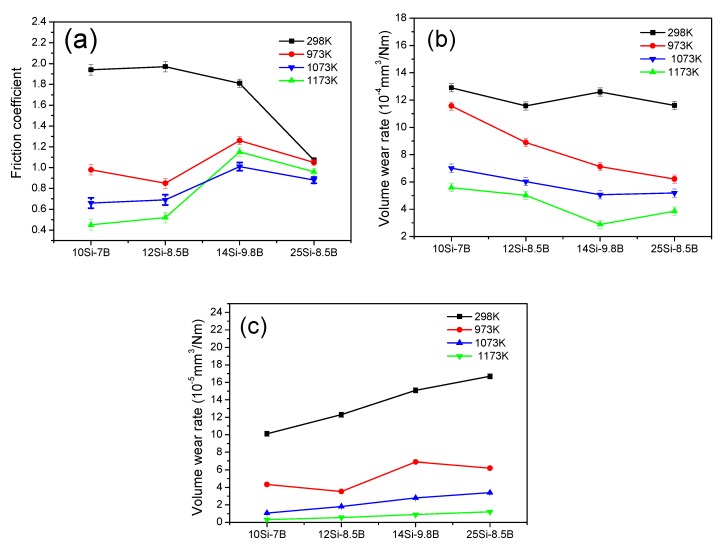
(**a**) Friction coefficients and (**b**) volume wear rates of the plate specimens of Mo-Si-B alloy sliding against the Si_3_N_4_ ball specimens (**c**) in the temperature range of 298 to 1173 K.

**Figure 6 materials-12-02011-f006:**
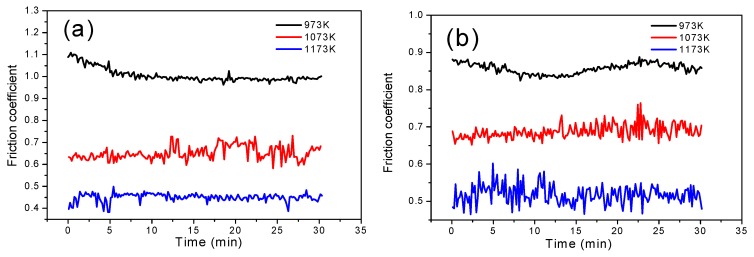
Friction coefficients of (**a**) 10Si-7B, (**b**) 12Si-8.5B, (**c**) 14Si-9.8B, and (**d**) 25Si-8.5B alloy doped with 0.3 wt % La_2_O_3_ plate specimens sliding against Si_3_N_4_ ball specimens in the temperature range of 973 to 1273 K.

**Figure 7 materials-12-02011-f007:**
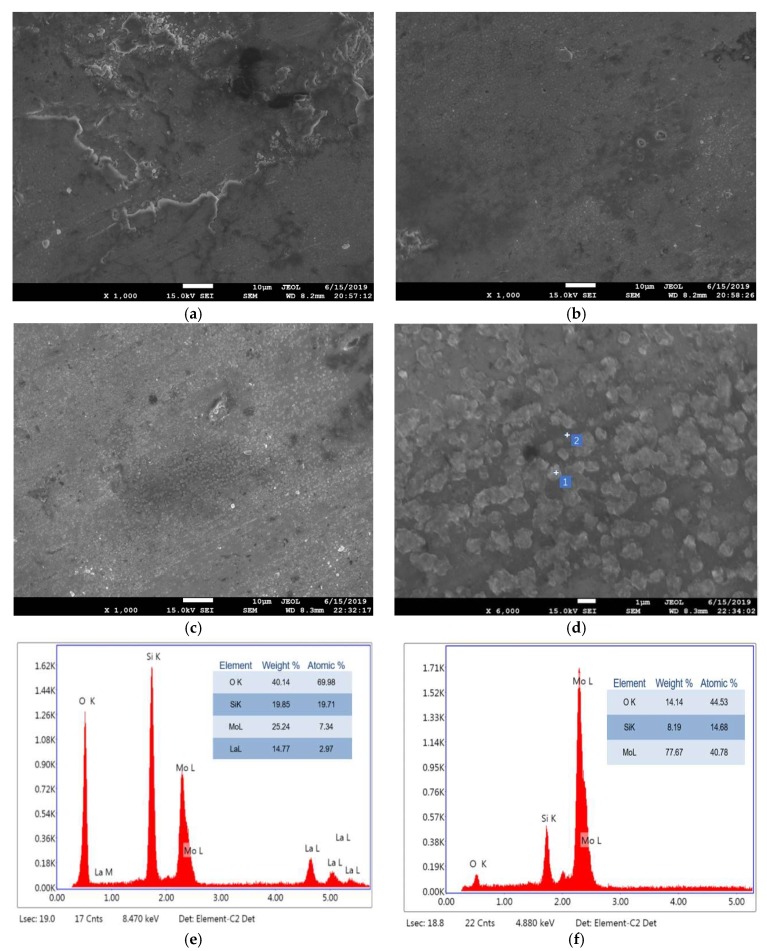
Representative SEM micrographs of 10Si-7B alloy after friction test: (**a**) at 973 K, (**b**) at 1073 K, (**c**) and (**d**) at 1173 K. (**e**,**f**) the EDS spectrums of P1and P2 in (**d**), respectively.

**Figure 8 materials-12-02011-f008:**
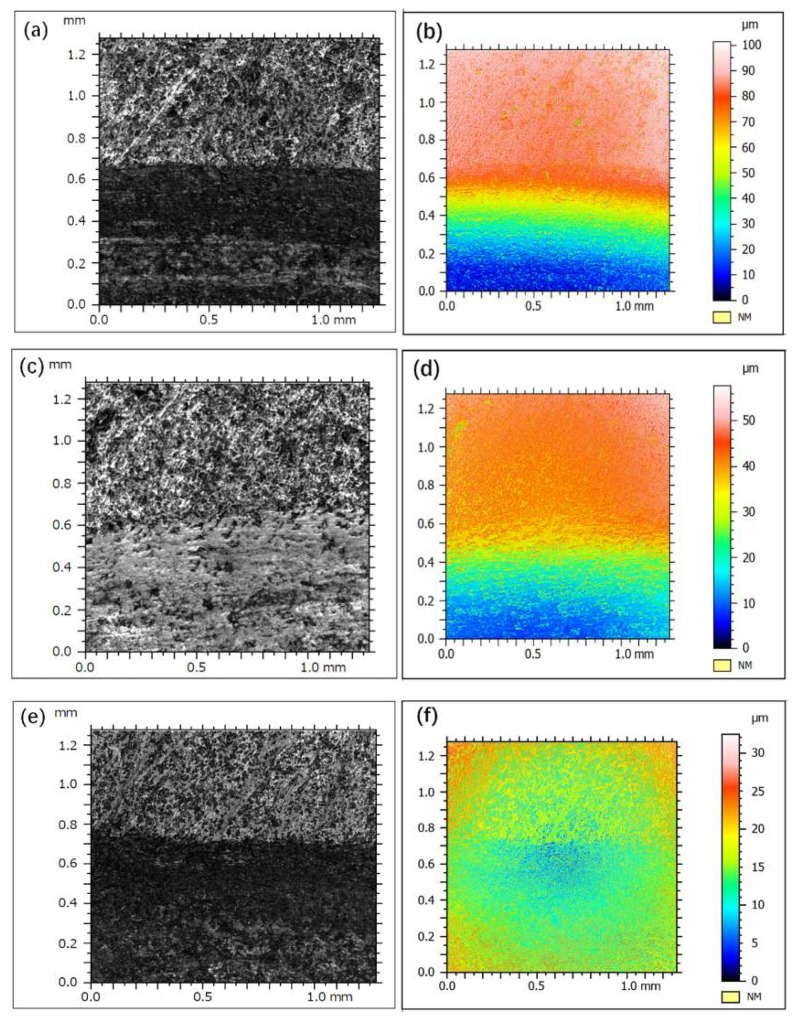
Confocal microscope images of Mo-Si-B alloy doped with 0.3 wt % La_2_O_3_: (**a**,**b**,**i**) 10Si-7B, (**c**,**d**,**j**) 12Si-8.5B, (**e**,**f**,**k**) 14Si-9.8B, and (**g**,**h**,**l**) 25Si-8.5B.

**Table 1 materials-12-02011-t001:** Compositions of four series of alloys with different Si and B contents (wt %).

Alloys	La_2_O_3_-Mo	Si	B
10Si-7B	96.00	3.38	0.91
12Si-8.5B	94.96	4.18	1.14
14Si-9.8B	93.89	5.03	1.36
25Si-8.5B	89.20	9.79	1.28

**Table 2 materials-12-02011-t002:** Size and mechanical properties of Si_3_N_4_ ceramic ball.

Diameter (mm)	Surface-Roughness (μm)	Density (g/cm^3^)	Vickers-hardness (GPa)	Fracture-Toughness (MPa·m^1/2^)	Modulus of Elasticity (GPa)	Poisson Ratio
6	0.1	3.15	32.6	6.5	304	0.288

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
