# Peer review of "Tribological Properties of Mo-Si-B Alloys Doped with La2O3 and Tested at 293–1173 K"

_materials, 2019, doi:10.3390/ma12122011_

Round 1
Reviewer 1 Report
The manuscript presents interesting research results, however, it requires amendments in order to raise its scientific level.
It is necessary to place error bars on the charts.
In the manuscript, the results of the research are discussed only. It is necessary to supplement with a deeper analysis of the obtained results.
Author Response
Thanks for your comments concerning our manuscript. Those comments are all valuable and very helpful for revising and improving our paper, as well as the important guiding significance to our researches. Our article has been edited using Editsprings’s Language Editing Express Services. We have studied comments carefully and have made correction which we hope meet with approval. Revised portion are marked in red in the paper. The responds to the reviewer’s comments are in the attachment.

Reviewer 2 Report
General:
The paper deals with tribological and mechanical properties of Mo-Si-B alloys doped with La2O3 at various temperatures. The paper is interesting and original; however, has some drawbacks in terms of presentation of the results and the clarity of writing. Although the language is grammatically mostly correct, the sentences are often over-complicated and sometimes difficult to follow. Some content related issues were also observed. Detailed suggestions for the improvement of the manuscript are provided below.
Detailed suggestions for the improvement of the manuscript (numbers refer to corresponding lines)
General observations:
- Title: The title should be improved to better encompass the actual content of the paper to e. g. “Tribological properties of Mo-Si-B alloys doped with La2O3 and tested at 293-1173 K”.
- 65: Authors say that “NH4La(Mo2O7)2 was synthesized by slow adding of La(NO3)3 to the above solution”; however it is not clear to which “above solution” the authors refer. If authors refer to (NH4)2Mo2O7, they should in the previous sentence on line 64 point out that this is a solution since it is currently not clear because crystallization was applied.
- 71: Please describe, how many samples were produced possibly by using a table) and provide their (possibly short) denotations which should then be used throughout the entire manuscript.
- 79: Please provide information on the producer of the HT-BPT.
- 87-95: It is not completely clear, how the tribological test set-up was and what the shape of the wear is. Please, describe the test set-up in more detail before describing the determination method for the evaluation of the wear track volume.
- 90-92: The sentence is unclear (why did the extent of dispersion determine the calculation accuracy?). Please rewrite.
- 104-105: It would be better to use shorter names for the denotation of the samples since currently, they are difficult to follow. For example, the parts which are the same for all samples could be omitted and only the parts which are different could be used, e.g. 10Si-7B, 12Si-8.5B, etc. This refers to the entire manuscript where sample names are used.
- 108: Authors claim that with XRD on 25Si-8B, apart from Mo3Si and Mo5SiB2 no other phases were observed; however in Fig. 1, on 25Si-8B, alpha-Mo phase is also indicated with 2 peaks.
- Fig. 1: It would be useful if e. g. dotted lines were used to denote the positions of phases a-d (see attached Figure).
- 113: Authors claim that “surface energy of the mixed powders was increased after MA”. Please, explain on which basis this was concluded.
- 124: Please define here what are “inter-metallic compounds” or to which species this refers to. Now the definition is provided only at line 142.
- 146: What do the authors mean by “squeezed heads”?
- 153-155: This sentence (“When the load was…”) should be at the end of the paragraph, on line 161. At the same time, also coefficients of friction at other loads should be described.
- 174: For the room temperature, it is suggested to use either “298” or “room” consistently throughout the manuscript in order to avoid confusion.
- 182: Authors claim that friction first decreased and then increased in the temperature range of 298-1173 K; however, this is not in accordance with the results in Fig. 5a, where friction decreased with temperature in the range of 298-1073 K and increased first at 1173 K.
- 190-192: Authors claim that “the oxide film produced at high temperatures played a significant role in lubrication”. The occurrence of oxidation should be supported by empirical evidence such as e. g. EDX, or similar.
- 219-223: This explanation is not in accordance with the results in Fig. 6, since at 1073 K friction for 14-Si and 25-Si were lower than for 10-Si and 12-Si.
- 209-229: For such statements, empirical evidence such as EDX - proving the formation of oxides - is needed.
- Fig. 6 caption: Describe what is presented in Figures (a) - (d).
- Fig. 7: Maybe for images (a, c, e, g, I, j, k, l) it would be better to use less contrast or to use another color than black-white because due to a significant amount of black, it is currently difficult to recognize specific features in these images.
- 264-266: In order to support this statement, empirical evidence is needed – see comments to lines 190-192 and 209-229.
- 255: Instead of “Wear pattern, contrast and digital composite images” it would be better to use only “Confocal microscope images”.
Grammar and writing issues:
- 16: Instead of “were grown up with enhancing the test load”, it should be “increased with the test load”.
- 40: It should be “34.9 and 390.4 GPa, respectively.”
- 43: It should be “but also covers”.
- 76: It should be “a holding time”.
- 79: It should be “a high temperature”.
- 81: It should be “testing conditions were:”
- 87: It should be “a grinding mark without a regular spherical morphology”
- 88: Instead of “dual friction”, it would be better to say “friction test”.
- 107: Instead of “composition”, it should be “components”.
- 107: It should be “...alloy were Mo3Si and…”
- It is not necessary to put a comma immediately after “since”. This applies to the entire manuscript.
- 118-120: Please rephrase the sentence. Currently, it could be falsely interpreted that the black and grey phases were present only in 10Si-7B and 12Si-8.5B.
- 121: Instead of “enhancing”, it should be “increasing”.
- 122 and 180: There is an additional unnecessary point at the end of the sentence.
- 125: It should be “...alloy was comprised almost of total...”.
- 126: Instead of “the least”, it should be “the minimum”.
- 137 and 138: It should be “doping with La2O3”.
- 142: Instead of “was grown up”, it should be “increased”.
- 153: Instead of “increment”, it should be “increase”.
- 155-156: It should be “The reason was the increased contact surface…”
- 158: Instead of “grinding materials”, it would be better to use “sliding materials”. This applies to the entire manuscript.
- 158: There is no need to use “-” between “two” and “friction”.
- 159: Instead of “enhancing”, it should be “increasing”.
- 159: Instead of “micro convex body on the surface”, it should be “surface asperities”.
- 163: It should be “The load affected…”
- 166: “by increasing the load” should be omitted since this is already clear from the context.
- 167-168: It should be “more La2O3 particles were detached from the alloy which increased 3-body abrasion.”
- 171: “wear” should not be capitalized.
- 173: There is no need to use “-” between “friction” and “coefficients”. This applies to the entire manuscript.
- 176: It is more accurate to say “coefficients of friction decreased”, i.e. without “were”. This applies to the entire manuscript, e.g. line 181, 190, etc.
- 184: The sentence should begin with: “For Mo-14Si-9.8B-0.3La2O3 alloy the lowest…”
- 185: The sentence should begin with: “For Mo-25Si-8.5B alloy, the friction coefficients…”
- 188: It should be “the volume wear rates”
- 190: The sentence should end with: “decreased temperature and Si and Mo content.”
- 198: Instead of “large”, it should be “high”.
- 199: Instead of “friction temperature”, it should be only “temperature”.
- 200: Instead of “immediate”, it should be “initial”, instead of “but”, it should be “and at the same time”.
- 201: Instead of “less”, it should be “lower”.
- 201-202: The sentence should begin with: “At 1173 K, the lowest average friction coefficients were obtained for…”
- 203-204: It should be “However, for Mo-14Si-9.8B-0.3La2O3 and Mo-25Si-8.5B--0.3La2O3 the average friction coefficients…”
- 205: It should be “fluctuated”.
- 210: Instead of “molybdic”, it should be “molybdenum”.
- 212: Space is missing between “MoO3” and “was”.
- 212-213: The sentence should be only “MoO3 acted as a lubricating film.”
- 216: It should be “When the test temperature was…”
- 219-220: It should be “is higher than the low content of SiO2 in Mo-14Si…”
- 228: It should be only “Fig. 7 shows the laser confocal images of the worn…”
- 232-233: Instead of “wear volume fraction”, it should be only “wear volume”.
- 234: It should be “According to the laser confocal images of the wear tracks from Fig. 7 (i-l), obvious furrows were observed on the surfaces of Mo-Si alloys, as are highlighted by arrows in Fig. 7.”
- 237: It should be “…in the parallel direction…”
- 239-240: Instead of “left furrows by wearing”, it should be “wear tracks”; and “obviously” could be omitted.
- 241: “they were” could be omitted; and instead of “as are highlighted”, it should be “as highlighted”.
- 245: It should be “…hardness is one of the main parameters…”
- 246: It should be “…of a material and in the present study the wear resistance…”; and instead of “increment”, it should be “increase”.
- 248: It should be “…hardness of materials are linearly related…”
- Fig. 7: Letters g, h, j, k, and l should be bold – the same as other letters in the Figure.
- 261: It should be “decreases”.
- 262 and 264: Instead of “enhancing”, it should be “increasing”.

Author Response

(The authors gave the same response as above.)

Round 2
Reviewer 1 Report
I accept the manuscript in its present form
Author Response
Dear Reviewer,
We appreciate for Reviewer’s warm work earnestly, and hope that the correction will meet with approval.
Once again, thank you very much for your comments and suggestions.
Reviewer 2 Report
The authors have addressed all the issues raised by the reviewer and have carefully conducted the required corrections. From the reviewer's point of view the paper is now suitable for publication.
There is however, one one minor issue which authors have overlooked and would be useful to address in the final version of the paper:
- In their response to the comment on line 113 regarding the surface energy of particles, authors have explained how they came to their respective conclusion, but haven't included their explanation into the manuscript. It is suggested to add the mentioned explanation into the manuscript since it is useful for understanding of the described mechanical alloying (MA) concept.
Author Response
Dear Reviewer,
Thank you very much for pointing this important point. I have added the mentioned explanation into the manuscript.
We appreciate for Reviewer’s warm work earnestly, and hope that the correction will meet with approval.
Once again, thank you very much for your comments and suggestions.